# Estimating the Cost of Internet Censorship in China: Evidence from a Gamified Remote Platform

**Jijian Fan** [1,2,*] and **Runquan Guan** [3]

1    School of Economics, Zhejiang University, Hangzhou 310058, China
2    Institute for Fiscal Big-Data and Policy, Zhejiang University, Hangzhou 310058, China
3    Department of Astromony and Astrophysics, University of California, Santa Cruz, CA 95064, USA
*    Correspondence: jijianfan@zju.edu.cn

**Abstract:** We exploit internet censorship intensity changes due to political events to study the impact of internet censorship on online laboor work in China. With a unique dataset from the Ingress (video game) community platform, a difference-in-differences design shows that an increase in China's internet censorship intensity during politically sensitive dates, while not affecting the amount of volunteer working time, reduces online labour work efficiency by eight percent for volunteers from mainland China relative to those elsewhere. This efficiency loss due to internet censorship can be a proxy for the labour productivity loss for Chinese oversea e-commercers, freelancers and other related online workers.

**Keywords:** internet censorship; Great Firewall; online work; Ingress (video game)

**JEL Classification:** L86; J22

## 1. Introduction

Many countries have implemented specific ways of internet censorship: some countries implement selective blocking during wartime, others may keep censoring more routinely. Internet censorship strengthens the cybersecurity (Hassib and Shires 2021) and authoritarian regimes (Lawson and Lawson 2002), brings political stability for the authorities (Lorentzen 2014) and local protectionism for domestic firms (Qiu 1999; Xu and Albert 2014), but is at the cost of freedom and human rights (Zarwan 2005). The Great Firewall of mainland China (GFW) is a series of legislative actions and internet technologies to regulate internet access in China. The GFW has played an essential role in Chinese internet censorship by blocking foreign websites and slowing down specific internet traffic. The censorship brings some benefits to the country: in terms of politics, it keeps out unwanted ideologies and maintains political stability; in terms of economy, it results in local protection for the domestic internet industry (Chen and Yang 2019), especially state-owned firms (Xu and Albert 2014).

Apart from the human rights criticism, censorship may also dampen the economic productivity by segregating the cyberspaces and stopping information flows (Qiu 1999). One of the economic losses is domestic people's inability to access foreign websites and online services, which may create information friction and harm labour productivity. For example, without bypassing GFW, a software engineer may not access commonly used codebases such as GitHub and Stack Overflow, and export retailers cannot advertise their products using Google Ads. Even if the censorship only increases the latency of internet access, it may still significantly distort the labour market given the attention is scarce (Bartoš et al. 2016). It is essential to know the economic cost while implementing such censorship.

However, measuring the causality between China's internet censorship and labour outcome is difficult due to several reasons. First, the Chinese government is reluctant to disclose the details of GFW censorship, such as what will be censored and when it will

trigger (King et al. 2013; Yang and Liu 2014). Second, it is hard to find a record of labour production for censorship-related studies, especially when firms face the law . Third, even if we acquired such a dataset, it is hard to relate the impact on labour productivity to the changes in censorship intensity. We do not know how censored contents are used in labour production, so it is unlikely to find a corresponding counterfactual. Fourth, while the benefit of accessing uncensored internet may take time to realize, short acquisition of uncensored information does not change people's internet usage patterns (Chen and Yang 2019). Therefore, while a vast wealth of literature, such as Ding et al. (2018) and Hassid (2020), has studied the impact of media censorship on the financial markets due to information frictions, few studies have focus on the direct effects of censorship on labour production outcomes.

We acquire a novel dataset from a video game community that records online volunteer jobs. Game players serve as long-term volunteers to do simple and repetitive work online: they assess whether a given place-of-interest satisfies specific criteria, further refining the game content. To browse the website and make the assessment, one needs to access the Google Account Service and Google Maps Service, which the GFW censors. We exploit the changes in censorship intensity due to political events to study the effect on such assessment jobs. By comparing the performance between volunteers inside and outside mainland China, a difference-in-differences design shows that experienced GFW bypassers still have access to restricted services during the periods with increased internet censorship. However, while the amount of time spent on volunteer work is unchanged, an increase in censorship intensity leads to an eight percent reduction in working efficiency. Given that Ingress players are familiar with bypassing the GFW, our result implies a lower bound of working efficiency loss for any labour productivity subject to internet censorship in China.

This paper contributes to the literature by addressing the link between internet censorship and its direct impact on the time and efficiency of online labour work. The estimates offer a lower bound of productivity loss from internet censorship. The study on the unpaid, recreation-oriented job might be generalized in a much broader online labour production issue, such as software engineers reaching online codebases. We also estimate changes in both intensive and extensive margins, finding that the GFW can strengthen changes in the intensive margin.

The rest of this paper is organized as follows. Section 2 shows the background information of the online voluntary Ingress players and how it is related to GFW censorship. Section 3 describes the dataset. Section 4 details the empirical strategy. Section 5 presents the results, and Section 6 concludes our findings.

## 2. Background Information and Data

### 2.1. Ingress and the Online Voluntary Job

In November 2012, Niantic, Inc. launched its first location-based augmented reality (LBAR) mobile game, Ingress. Players use a mobile device to locate and interact with portals which are close to the player's real-world location. Portals are physical points of historical and cultural interest, such as statues, monuments, historic buildings, and other displays of human achievement, and are the core resource of the game.[1] Players join two factions and "battle" each other to control portals and make certain actions. The unique design of the game attracts millions of players around the world, and these players are highly intrinsically motivated to contribute to the game: they build communities, hold events, travel to other countries to play, organize global operations, and contribute to the game's content by designing in-game missions and suggesting new portals. For example, in 2016, an official event, "Aegis Nova", was held in Tokyo and over 15 thousand players from all around the world participated. In 2019, more than three hundred players in Japan, China, India, Oman, Australia, Singapore and Malaysia completed an operation that requires organized, simultaneous actions on more than a thousand portals. Such events illustrate the passion of players for the game.

There are more than 5 million portals in total: while Niantic made the initial portal network, most of the subsequent portals are crowd-sourced by players.[2] Experienced players are allowed to nominate and evaluate portal candidates through an online platform named "Niantic Wayfarer". This platform was initially called "Operation Portal Recon" for Ingress players only. Later on, Niantic developed more games based on the same portal database. Faccio and McConnell (2020) studied another LBAR game, Pokémon Go, which is more popular than Ingress. Figure 1 shows the user interface of "Niantic Wayfarer". Volunteers perform simple, repetitive work to rate portal candidates by whether they are culturally or socially valuable, safely accessible, visually unique, and are in the accurate location. Portal candidates receiving enough positive evaluations will become portals, otherwise they will be rejected.[3] An agreement is made when a volunteer's assessment matches the decision. As a reward, players will receive tiered in-game medals when they reach a certain number of agreements.

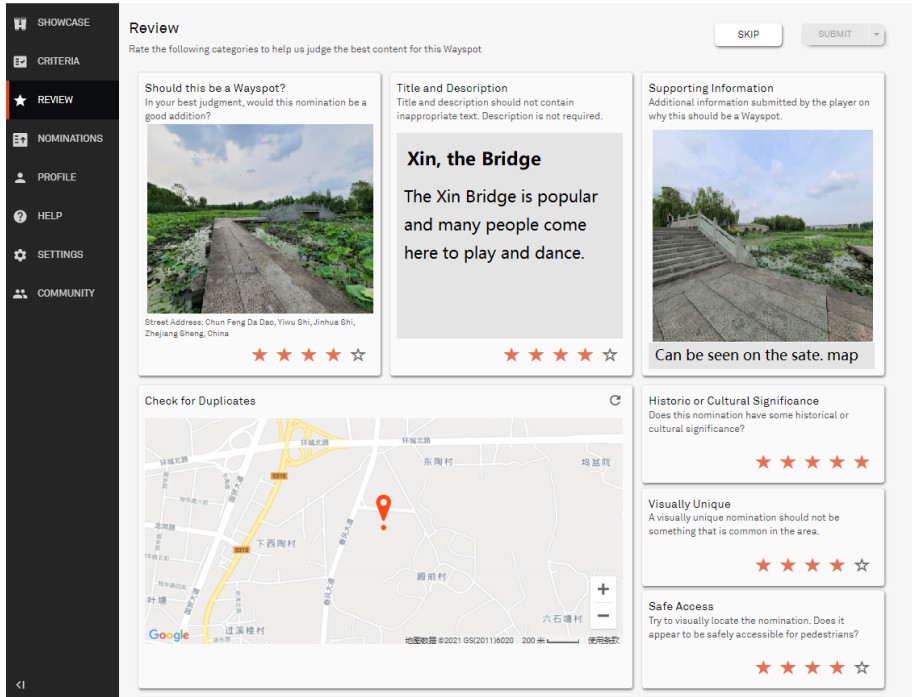

**Figure 1.** The user interface of the "Niantic Wayfarer". Experienced video game players who contribute as volunteers will review each nomination and judge whether it satisfies certain acceptance criteria by giving a score. Once the score is "close to" other volunteers' rating, an accept/reject decision will be made, and the volunteer receives one "agreement". The detailed criteria is confidential and thus is unknown to the players. Note that some of the original content in Chinese has been translated into English.

Similar to other parts of the world, Ingress is also popular in China. To encourage reviewers and make more portals available online, in April 2019, Shenzhen Enlightened, an Ingress player community in mainland China, organized a non-official volunteer service competition. Volunteers needed to complete the task of having no less than 100 new agreements for each weekly checkpoint. Every week their increments are checked and recorded. If the increments are less than 100, the participant is out. A typical review takes between 10 s to 3 min, and an experienced reviewer can make 50–70 percent of their reviews matching the decision. Thus, it may take 3 to 4 h to finish 100 agreements. However, since the increments are usually not counted right after the review is made, a reviewer may have to make more than 100 reviews per week to stay in the competition. Winners who survive all week are eligible for a raffle for a real-world money prize, souvenir, or redeemable in-game inventory codes at the end of the competition.

## 2.2. The Great Firewall

Players in mainland China have to bypass GFW to access Ingress and its portal review features. There are two main features of the GFW. First, the GFW works as a domain filter: Chinese government authorities block specific domains and IP addresses, most of which belong to foreign business, pornography, gambling, and media/news providers (Hoang et al. 2021). Ingress, which depends on Google Services, is also blocked. However, people can use encrypted internet traffic to bypass the domain filter, such as a virtual private network (VPN). Therefore, the GFW also introduces a second feature as a content filter: it first evaluates all internet data ("TCP packets") and assigns a score of being "abnormal" for each data traffic flow, then blocks all internet traffic with a score higher than a certain threshold (Mölsä 2005; Ptacek and Newsham 1998). This includes content from certain websites, with specific keywords, or particular characteristics unwanted by the government. Even if the government claims that only a tiny fraction of domains, IPs, and content are blocked, it still technically screens all internet traffic to distinguish the unwanted ones. On the one hand, this screening may slow internet traffic; on the other hand, the detection process faces the classic statistical trade-off between Type-I and Type-II errors. The GFW uses a particular algorithm to detect suspicious internet traffic flow as all data packages are encrypted. The technical detail of detecting and blocking is unknown, treated as a national secret. However, the authority may change the threshold to control the intensity of internet censorship. A lower threshold blocks more abnormal internet traffic but leads to a high false-positive rate: normal internet usage, such as browsing an overseas university website or using a foreign online software, will also be impacted. Note that internet censorship also includes the regulation of domestic internet content, which is not in the interest of this study.

The government authority faces a trade-off between successfully blocking unwanted information and letting regular internet traffic through. Therefore, when the government values political stability over economic productivity, the intensity of internet censorship will be higher. The Ingress volunteer competition lasts for more than half a year. It overlaps with several political events, including Tiananmen Square Protest Memorial (4 June), National Cybersecurity Week (16–22 September), National Day (1 October), and CCP National Meeting in 2019 (28–31 October), during which the intensity of internet blocking is temporarily raised. Such political events are less likely to affect people's daily life in the short term. On the one hand, the Tiananmen Square Protest Memorial in practice is not allowed to be memorized, or even publicly discussed in China, and thus the Chinese people do not organize memorial events. On the other hand, political meetings and events are less relevant to people's physical activities and thus their behaviour is less restricted.[4] The government simply raises the intensity of censorship as a preventive measure for political instability.

## 2.3. Censorship and the Ingress Gameplay Friction

For Ingress players in China, their access to the game relies on the functional stability of their VPN, and thus the internet friction may affect their connection. The outdoor gameplay is negligibly affected as only the log-in process requires bypassing the censored connection, once successfully logged in the operational internet communications are not censored. However, the reviewing work requires continual connection to the censored Google Map services. Therefore, the intensified censorship, due to the politically sensitive periods, provides an exogenous shock in internet friction when accessing the reviewing service.

Notably, this online volunteer competition is held within the Chinese Ingress player community: all Chinese participants share similar cultural backgrounds. However, due to foreign education, working abroad, or travelling, some participants were physically outside mainland China during the competition and thus were not subject to the GFW censorship. Playing Ingress leaves digital traces that contain geographical information. For example, if a player interacts with a portal whose real-world location is in New York, the player must be physically present in New York at that time. We exploit these temporarily

out-of-censorship populations as the counterfactual and study the effect of strengthening the GFW censorship.

While there is still a dichotomy between work and leisure in an economic context (Goux et al. 2014; Guryan et al. 2008), a vast wealth of literature has noted that the distinction between playing and working becomes more and more blurred: fans may contribute persistently to their beloved subculture, just as workers contribute to their serious jobs (Busse 2015; Goggin 2011; Kücklich 2005; Taylor et al. 2015). Given that Ingress has only several thousand active players in China, all of which are highly loyal, we used volunteer work as a proxy for internet-related labour work. The comparison between the work performance of Chinese players inside and outside China leads to much less bias than simply comparing Chinese and foreign players.

## 3. Data

We obtain the complete record from the Shenzhen Enlightened player community with the number of weekly agreements (both self-reported and spot-checked numbers) for each player in the volunteer competition. Two hundred and eleven players served as volunteer reviewers and made 3229 observations over 30 weeks. As playing Ingress leaves digital, geographical traces, we made two waves of spot checks during the first week of June and September by manually retrieving each player's most recently played location. If a player was outside mainland China during at least one spot, their were considered GFW-unrestricted. By checking players' locations we identified 180 players in mainland China subjected to GFW-restricted access, and the remaining 31 players were unrestricted. To further reveal the impact of the GFW on the volunteers' behaviour, we also acquired the result of a questionnaire survey from the organizers. The survey asked how many hours a participant would spend reviewing, helping us identify the voluntary labour input.

Table 1 shows the summary statistics of the performances between volunteers inside and outside mainland China. Panel A shows the reviewing performance: there is no significant difference between volunteers inside mainland China or outside in terms of the weekly agreement number or average number of weeks survived. The fact that the event is held within the Chinese player community shows that most participants are in mainland China, resulting in a relatively large sample size (85 percent). Panel B shows the weekly hours spent reviewing. The data are obtained from the voluntary questionnaire survey; therefore, the reply rate is limited. However, the comparison shows that participants in mainland China typically spend 3.88 h per week volunteering, much more than those outside China, spending only 2.37 h per week.

**Table 1.** Summary Statistics.

| Location | Mainland China | Other |
|---|---|---|
| GFW Censorship | Restricted Access | Unrestricted Access |
| Panel A: Review Records | | |
| Number of Weekly Agreements | 179.73 | 173.51 |
| (Self-reported) | (158.38) | (125.27) |
| Number of Weekly Agreements | 178.94 | 172.58 |
| (Spot-checked) | (157.82) | (126.38) |
| Average Survival Weeks | 12.4 | 11.09 |
| | (10.04) | (8.45) |
| Number of Volunteers | 180 | 31 |
| % | 85% | 15% |
| Number of People Survived (%) | | |
| Week 9 | 115 (64%) | 18 (58%) |
| Week 26 | 65 (26%) | 7 (23%) |
| Week 30 (Completion) | 45 (25%) | 6 (19%) |
| Number of Observations | 2813 | 416 |

**Table 1.** *Cont.*

| Panel B: Questionnaire Survey | | |
|---|---|---|
| Number of Replies | 66 | 14 |
| Event Completion | 33 | 5 |
| Weekly Hours Spent Reviewing | | |
| Average During Event | 3.88 | 2.37 |
| | (2.17) | (1.74) |
| Maximum During Event | 6.03 | 4.71 |
| | (2.23) | (2.51) |
| Average After Event | 1.65 | 1.14 |
| | (1.33) | (0.53) |

Note: This table shows the summary statistics of the review performance and average weekly volunteering hours for participants from mainland China and other regions. Standard deviations are in parentheses. The number of agreements for comparison is collected from non-treatment periods, i.e., weeks 1–6, 9–21, 26–27, and 30.

## 4. Empirical Design

The nature of this volunteer competition provide us with a group difference on whether volunteers are subject to the GFW censorship, and an exogenous time difference in internet blocking intensity. We exploit the fact that internet blocking is strengthened during certain weeks and use Chinese-speaking players who live outside mainland China as the counterfactual, assuming that players inside and outside mainland China have similar playing patterns except for GWF internet traffic blocking, and implement a difference-in-differences design by estimating the following equation:

$$y_{i,t} = \beta Restricted_i \times Sensitive_t + \mu_i + \nu_t + \varepsilon_{i,t} \tag{1}$$

where $y_{i,t}$ represents the number of agreements of volunteer $i$ in week $t$. We use both the self-reported and spot-checked outcomes. The dummy variable $Restricted = 1$ if the volunteer agent is in mainland China, i.e., access restricted by the GFW internet censorship, and $Restricted = 0$ if the agent is outside mainland China. $Sensitive = 1$ if the checkpoint week overlaps with the GFW strengthening period (weeks 7–8, 22–25, 28–29), otherwise $Sensitive = 0$. Note that the variables $Restricted$, $Sensitive$ and constant terms absorbed by the individual and time dummies, $\mu_i$ and $\nu_t$, respectively. The coefficient $\beta$ represents the effect of GFW strengthening on the number of volunteer agreements with restricted access, compared to those with unrestricted access. $\beta < 0$ is expected if GFW strengthening reduces online assessment work. Despite the fact that a review may become an agreement several days after it was made, we also estimate the effect using the lagged treatment weeks, i.e., estimate the following equation:

$$y_{i,t} = \beta Restricted_i \times Sensitive_{t-1} + \mu_i + \nu_t + \varepsilon_{i,t} \tag{2}$$

where the coefficient $\beta$ has similar interpretation, but allows for a longer realization time. In addition, individual and week-fixed effects are controlled in all regressions.

Due to the nature of the volunteer competition and our dataset, a volunteer who fail to have 100 weekly increments at the checkpoint is disqualified. Therefore, we can no longer observe this volunteer's number of agreements. The intensified internet censorship may increase the probability of participants quitting instead of decreasing the number of agreements. Quitting is more likely when the blocking effect is vital: in an extreme case where the internet service is entirely cut, the number of agreements should fall to zero such that all affected individuals quit. In such a case, as we cannot observe the outcomes of volunteers who dropped out, Equations (1) and (2) cannot capture the actual treatment effect. To rule out this possibility, we also estimated a survival model using a Cox proportional hazard model (Cox 1972) as follows:

$$h_i(t) = h_0(t) \times \exp\left(\beta_1 Restricted_i \times Sensitive_t + \beta_2 Sensitive_t + \beta_3 Restricted_i + \varepsilon_{i,t}\right) \tag{3}$$

where $h_i(t)$ represents the hazard function of volunteer $i$ at time $t$. We assume that each individual has a constant baseline probability of quitting each week and that these probabilities may increase during politically sensitive weeks for volunteers inside and outside mainland China. The exponential of $\beta_1$ will capture the hazard ratio of GFW strengthening for mainland China volunteers relative to overseas Chinese players. $\beta_1 < 0$ is expected if GFW strengthening increases the probability of quitting.

To unveil whether the potential changes in performance are due to changes in productivity or reduced efforts of volunteering, we estimate the effect of censorship on the amount of time used per week. Ideally, we would like to use the same difference-in-differences design with each participant's weekly time as the outcome variable. However, unfortunately, as the questionnaire was not designed for our study, and the dataset does not have the individual-by-week record of weekly hours. Meanwhile, as the survey was anonymous, we cannot match the replies to a participant's performance. After talking with the organizers, we realized did not intended on asking the participants for their personal information, as GFW bypassing is illegal in China. They designed the questionnaire to ensure that answering the questions would not leak any identifiable personal information. Therefore, we changed our identification strategy: from the questionnaire we acquired each participant's average and maximal time spent reviewing per week during the competition and the average time reviewing after the competition (see Appendix A questions 2–7). We estimate the following equation:

$$y_i = \beta Restricted_i + \gamma Complete_i + \varepsilon_i \tag{4}$$

where the dummy variable $Restricted_i$ is set to 1 if the participants used a VPN service provided by themselves, their friends, or any third-party sellers, and 0 if they did not need to use a VPN to access the reviewing website. $Complete_i$ is a self-reported control variable indicating whether the participant finished the whole event. The coefficient $\beta$ estimates the difference in time spent between censored and uncensored participants. We are interested in the coefficients associated with different outcome variables $y_i$: (1) the average time spent reviewing after the competition, (2) the average time spent reviewing per week during the competition, and (3) the maximal time spent reviewing per week during the competition. The coefficient in (1) gives a benchmark to see how players inside and outside mainland China naturally differ in the willingness to volunteer. The difference between $\beta$s in (2) and (1) shows whether players in mainland China need to spend more time to fulfill the required 100 agreements per week. Finally, we assume that the weeks with the most time spent reviewing are likely associated with periods of strengthened GFW censorship. The difference between $\beta$s in (3) and (2) indicates whether players in mainland China are more likely to spend more time reviewing during the politically sensitive periods. Such a comparison will shed light on the changes in volunteering time.

## 5. Results

### 5.1. Effects on Work Performance

We first estimate the effect of strengthened internet censorship on the volunteers' performance, that is, the number of weekly reviews. Figures 2 and 3 show the number of self-reported and spot-checked agreements per week for volunteers inside and outside mainland China, respectively. In the shaded area, corresponding to the GFW strengthening periods, restricted volunteers performed worse than unrestricted volunteers. However, the weekly performance has relatively large random variations for both groups due to the limited sample size. Another interference comes from adaptive behaviour: if a participant in China anticipates the difficulty of intensified censorship, they may accumulate more agreements in advance. Therefore, we cannot see an obvious trajectory. The formal result shown in Table 2 shows the estimates of Equations (1) and (2). In Panel A, columns (1) and (3) show that compared with volunteers with unrestricted access, the GFW strengthening reduces the weekly number of self-reported and spot-checked agreements by 19.5 and 18.4, respectively, while these estimates are not statistically significant, columns (2) and (4) show that with lagged

treatment, the estimates of the average treatment effect increase both in magnitude (24.3) and significance. The results indicate a lagging impact of censorship: while an anticipated player can review more agreements before the censorship begins, all agreements are likely to be counted within a week, and after one week, a decline appears. This reduction corresponds to an 8–9 percent reduction in volunteering, as shown in Panel B.

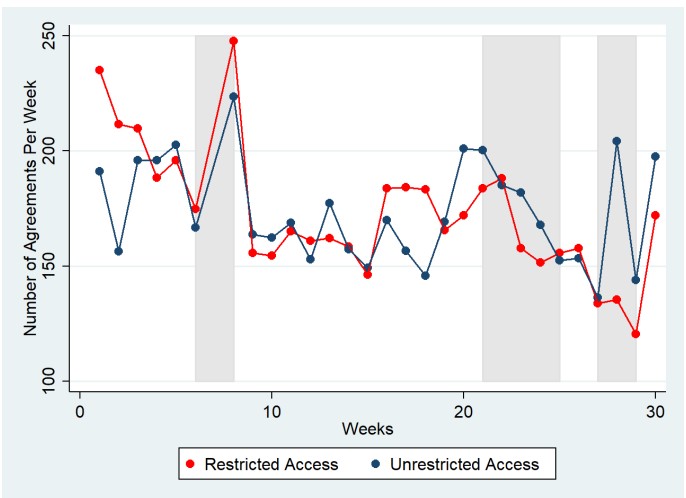

**Figure 2.** This figure shows the number of self-reported weekly agreements by volunteers inside and outside mainland China, i.e., whether they have restricted access. Shaded areas correspond to the GFW strengthening period for Tiananmen Protest Memorial (week 7–8), National Cybersecurity Week (week 22–23), National Day (week 24–25), and CCP's National Meeting (week 28–29). Note that the organization group intended to skip the checkpoint in week 7, anticipating GFW strengthening. The volunteers' online reviewing performance of the restricted group was lower than that of the unrestricted group, showing that the strengthening internet censorship decreased access and reduced the labour work outcomes.

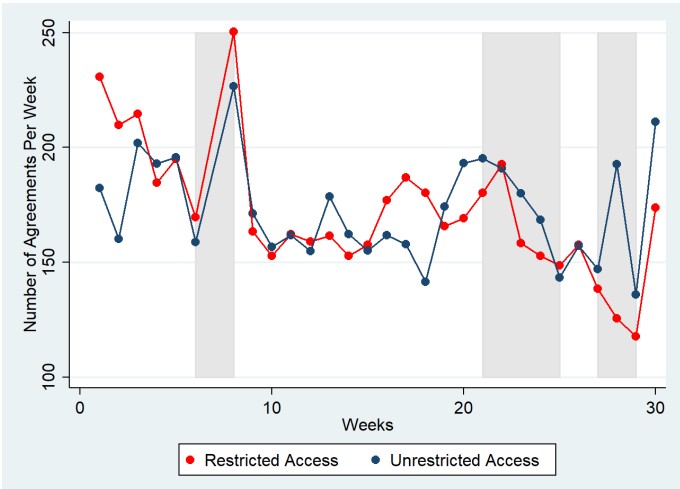

**Figure 3.** This figure shows the number of spot-checked weekly agreements by volunteers inside and outside mainland China, i.e., whether they have restricted access. Shaded areas correspond to the GFW strengthening period for Tiananmen Protest Memorial (week 7–8), National Cybersecurity Week (week 22–23), National Day (week 24–25), and CCP's National Meeting (week 28–29). Note that the organization group intended to skip the checkpoint in week 7, anticipating the GFW strengthening. The volunteers' online reviewing performance of the restricted group was lower than that of the unrestricted group, showing that the strengthening internet censorship decreased access and reduced labour work outcomes.

**Table 2.** Estimation of weekly agreements.

| | (1) | (2) | (3) | (4) |
|---|---|---|---|---|
| Panel A<br>Dep. Variable: Number of Agreements | Self-Reported | | Spot-Checked | |
| $Restricted_i \times Sensitive_t$ | −19.50<br>(12.13) | | −18.44<br>(11.86) | |
| $Restricted_i \times Sensitive_{t-1}$ | | −24.32 **<br>(10.43) | | −24.25 **<br>(10.63) |
| Mean Dependent Var. | 178.56 | 175.54 | 177.73 | 174.55 |
| Individual Fixed Effect | X | X | X | X |
| Time (Week) Fixed Effect | X | X | X | X |
| Observations | 3229 | 3087 | 3228 | 3086 |
| $R^2$ | 0.123 | 0.110 | 0.124 | 0.109 |
| Panel B<br>Dep. Variable: Natural Log of Agreements Number | Self-Reported | | Spot-Checked | |
| $Restricted_i \times Sensitive_t$ | −0.08 *<br>(0.04) | | −0.08 *<br>(0.04) | |
| $Restricted_i \times Sensitive_{t-1}$ | | −0.09 **<br>(0.04) | | −0.08 *<br>(0.05) |
| Mean Dependent Var. | 5.01 | 5.00 | 5.00 | 4.98 |
| Individual Fixed Effect | X | X | X | X |
| Time (Week) Fixed Effect | X | X | X | X |
| Observations | 3229 | 3087 | 3228 | 3086 |
| $R^2$ | 0.215 | 0.175 | 0.201 | 0.162 |

Note: Estimates show the effects of GFW strengthening on the number of weekly agreements. Panel A shows the effect on the magnitude, and Panel B shows the percentage effect using natural logs. Columns (1) and (2) use the self-reported outcomes, and columns (3) and (4) use results from spot checks. *Restricted* = 1 if the volunteer is in mainland China, i.e., suffers from the GFW, and *Restricted* = 0 if the volunteer is outside mainland China. *Sensitive* = 1 if that week is associated with GFW strengthening (weeks 7–8, 22–25, 28–29), otherwise *Sensitive* = 0. The individual-fixed effects and the quadratic form trend effect are controlled in all regressions. Standard errors are in parentheses. * $p < 0.10$, ** $p < 0.05$.

Notably, during the competition, the organizers realized the difficulty of accessing the uncensored internet for a significant share of participants in the first week of GFW strengthening and thus chose to skip the upcoming checkpoint in week 7, reducing the required increments from 200 to 150 in week 8. This adjustment made it more flexible for volunteers but effect the first wave estimate of GFW strengthening. To make the results more robust, we also show an estimate only using the observations after week 9, as shown in Table 3. The results imply that the average treatment effect is larger and more concentrated in the current week: the strengthened internet censorship reduces the number of weekly agreements by 27.73 (self-reported) and 24.84 (spot-checked), corresponding to a 12–13 percent reduction. The reduction effect is more substantial and statistically significant compared with the results from Table 2, indicating that GFW strengthening reduces the number of weekly agreements, and the postponed checkpoint gave the affected participants a break to catch up.

**Table 3.** Estimation of weekly agreements after week 9.

| Panel A | (1) | (2) | (3) | (4) |
|---|---|---|---|---|
| Dep. Variable: Number of Agreements | Self-Reported | | Spot-Checked | |
| $Restricted_i \times Sensitive_t$ | −27.73 ** | | −24.84 ** | |
| | (10.90) | | (10.78) | |
| $Restricted_i \times Sensitive_{t-1}$ | | −19.55 ** | | −19.65 ** |
| | | (7.62) | | (8.37) |
| Mean Dependent Var. | 162.70 | 162.70 | 162.22 | 162.22 |
| Individual Fixed Effect | X | X | X | X |
| Time (Week) Fixed Effect | X | X | X | X |
| Observations | 2057 | 2057 | 2056 | 2056 |
| $R^2$ | 0.066 | 0.065 | 0.067 | 0.067 |
| Panel B | | | | |
| Dep. Variable: Natural Log of Agreements Number | Self-Reported | | Spot-Checked | |
| $Restricted_i \times Sensitive_t$ | −0.13 ** | | −0.12 * | |
| | (0.06) | | (0.06) | |
| $Restricted_i \times Sensitive_{t-1}$ | | −0.10 *** | | −0.09 ** |
| | | (0.03) | | (0.04) |
| Mean Dependent Var. | 4.96 | 4.96 | 4.95 | 4.95 |
| Individual Fixed Effect | X | X | X | X |
| Time (Week) Fixed Effect | X | X | X | X |
| Observations | 2057 | 2057 | 2056 | 2056 |
| $R^2$ | 0.100 | 0.098 | 0.091 | 0.091 |

Note: Estimates show the effects of GFW strengthening on the number of weekly agreements using observations after week 9. Panel A shows the effect on the magnitudes, and Panel B shows the percentage effect using natural logs. Columns (1) and (2) use self-reported outcomes, and columns (3) and (4) use results from spot checks. *Restricted* = 1 if the volunteer is in mainland China, i.e., suffers from the GFW, and *Restricted* = 0 if the volunteer is outside mainland China. *Sensitive* = 1 if that week is associated with GFW strengthening (weeks 22–25, 28–29), otherwise *Sensitive* = 0. The individual-fixed effect and quadratic form trend effect are controlled in all regressions. Standard errors are in parentheses. * $p < 0.10$, ** $p < 0.05$, *** $p < 0.01$.

### 5.2. Effects on Participation and Working Time

The volunteer competition requires a minimum of 100 agreements per week to survive. This may lead to a sample selection bias: we do not observe the outcomes of those who drop out. However, if the censorship was solid and cut all international internet connections, the whole treatment group would immediately drop out, and our estimates would give no treatment effect. Therefore, we compare the survival rates between these player groups to rule out this case. Figure 4 shows the Kaplan–Meier estimates of the survival functions of volunteers with restricted internet access (inside mainland China) or without (outside mainland China). The estimate shows that participants from mainland China, those with restricted internet access, tend to have lower survival rates than those with uncensored internet access. However, the difference does not come from the GFW strengthening periods. Table 4 shows the Cox estimates of Equation (3). The coefficient estimate of $\beta_1$ is no different from 0, indicating that compared to those with unrestricted access, volunteers with restricted access do not present a higher hazard ratio of quitting due to not finishing the required 100 increments during the GFW strengthening periods. This rules out the possibility that the censorship was too strong such that volunteers in mainland China cannot work at all.

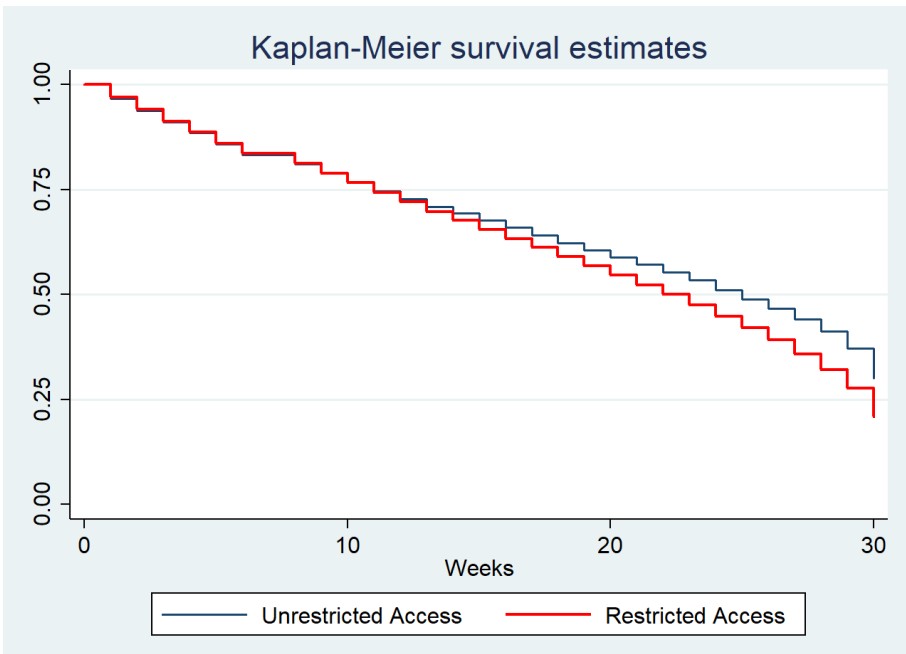

**Figure 4.** This figure shows the Kaplan–Meier estimates of the survival functions by volunteers inside and outside mainland China, i.e., whether they have restricted access. The estimate shows that participants from mainland China have lower survival rates compared with those with uncensored internet access, but the difference does not result from the GFW strengthening periods.

**Table 4.** Estimation of hazard rate.

| | (1) Hazard Ratio |
|---|---|
| *Restricted × Sensitive* | 0.12 |
| | (0.14) |
| *Restricted* | 0.13 ** |
| | (0.06) |
| *Sensitive* | −1.15 *** |
| | (0.13) |
| Number of Observations | 6417 |
| Number of Failures | 3178 |

Notes: Estimates show the impact of GFW strengthening on the hazard ratio. The table shows coefficients ($\beta$) and their standard errors. The relative risk ratio equals the exponential of coefficients ($\exp(\beta)$). *Restricted* $= 1$ if the volunteer is in mainland China, i.e., suffers from GFW, and *Restricted* $= 0$ if the volunteer is outside mainland China. *Sensitive* $= 1$ if that week is associated with GFW strengthening (weeks 7–8, 22–25, 28–29), otherwise *Sensitive* $= 0$. The hazard function is stratified by individual. Standard errors are in parentheses. ** $p < 0.05$, *** $p < 0.01$.

Table 5 shows the estimates of Equation (4). Column (1) shows that, without the competition, players on average spend 1.56 h every week performing voluntary portal reviewing. There is no significant difference between the players inside and outside mainland China, confirming that our research subjects are naturally similar in their time allocation between voluntary work and other things. Column (2) shows that during the event, participants, on average, spent 3.62 h per week reviewing. Notably, those in mainland China subjected to GFW censorship spent 1.48 h more per week than those who were not. Given that the voluntary labour outcome, i.e., the number of weekly agreements, is similar, this result implies that the existence of GFW censorship dampens online labour productivity. Finally, column (3) shows that the maximal amount of time spent by an access-restricted participant is 1.32 h more than that of an access-unrestricted participant, which is not statistically different from column (2), indicating that during a GFW strengthened period, players from mainland China Mainland did not increase or decrease their time spent reviewing. The results in columns (2) and (3) also rule out the possibility of the substitution effect: the strengthening of censorship may alternatively affect players' other

activities; for example, making their main jobs easier or harder. However, the null result implies that this is not the case: players in mainland China did not change their weekly time spent volunteering during the politically sensitive periods.

**Table 5.** Estimation of weekly time spent.

| Weekly Hours Spent Reviewing | (1) After Competition Average | (2) During Competition Average | (3) Maximum |
|---|---|---|---|
| Restricted | 0.52 | 1.48 ** | 1.32 * |
| | (0.37) | (0.63) | (0.68) |
| Complete | −0.10 | 0.23 | −0.03 |
| | (0.28) | (0.48) | (0.52) |
| Mean Dependent Var. | 1.56 | 3.62 | 5.80 |
| Observations | 80 | 80 | 80 |
| $R^2$ | 0.026 | 0.073 | 0.047 |

Note: Estimates show the effects of GFW strengthening on the weekly hours spent voluntarily reviewing. Column (1) uses the average weekly hours not in the competition, representing the difference between players' effort without the competition event. Column (1) uses the average weekly hours during the competition. Column (3) uses the maximal weekly hours during the competition, as a proxy for the hours spent during the GFW strengthening periods. *Restricted* = 1 if the volunteer is subjected to GFW censorship, otherwise *Restricted* = 0. *Complete* = 1 if the volunteer finishes all the requirements of the game, otherwise *Complete* = 0. Standard errors are in parentheses. * $p < 0.10$, ** $p < 0.05$.

### 5.3. Placebo Tests

To confirm that the estimated average treatment effect comes from the differentiated internet censoring effect during the strengthened periods rather than existing trends, other unobserved factors, or random coincidences, we implemented a series of Monte Carlo permutations as placebo tests (Dwass 1957). First, keeping the relative size fixed (180 treated and 31 control), we randomly labelled each individual to a treatment or control group and estimated the coefficients. With the randomly labelled treatment and control groups, the coefficient estimates should be distributed with zero means. Subsequently, we repeated the permutation 500 times and examined the null hypothesis to see if our estimated coefficients are statistically different from zero. Similarly, we randomly labelled the weeks as censorship strengthened to test whether our coefficient is different from zero.

Figures 5 and 6 show the result of permutation tests with randomly labelled treatment groups and treated weeks, respectively. We used the number of self-reported weekly agreements as the outcome variable. Our real coefficients (−19.50) in both the placebo tests were significantly different from zero ($p = 0.088$ and $0.001$, respectively), confirming that the estimated effect stems from internet censorship. We also used the spot-checked outcome for permutation tests, yielding a similar result and is thus not shown here.

### 5.4. Discussion

These results imply internet blocking harms online labour production primarily in the intensive margin: during these periods with exceptional internet traffic control, experienced internet users, if they previously know how to bypass the GFW, are likely to still be able to access uncensored resources. Our findings of reduced online laboyr work may be due to a longer latency, a more unstable connection, or a higher cost of finding software that provides censorship-bypassing services. However, we cannot rule out the impact on the extensive margin: the subjects in this study are experienced in internet censorship bypassing as they have played Ingress, requiring bypassing, for a while, yet for the general population, strengthened blocking may cut off their connections to the outside world. Therefore, our results show a lower bound for the impact of internet censorship on online labour production.

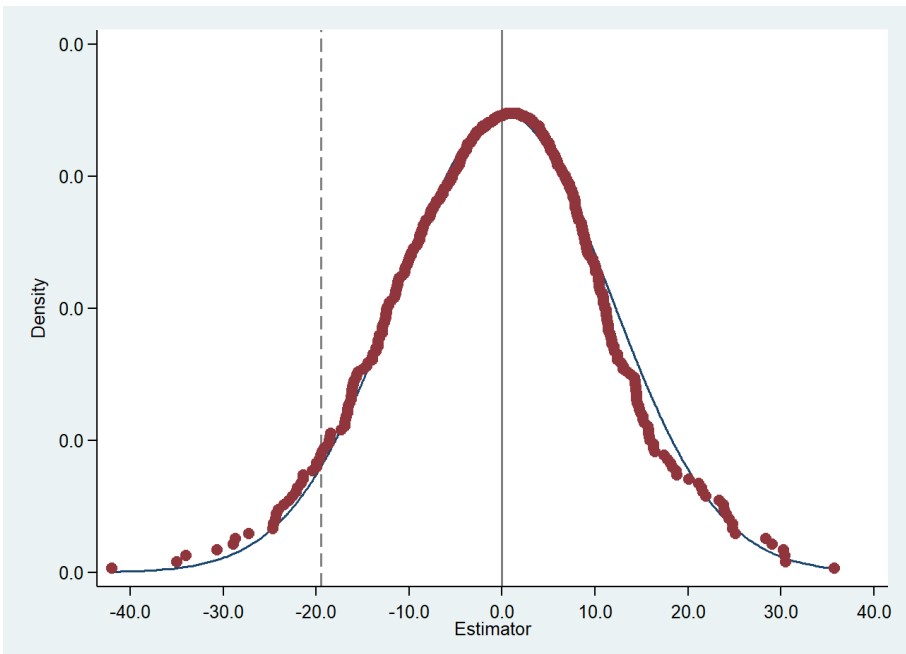

**Figure 5.** This figure shows the distribution of the estimated coefficients with the randomly permuted treatment assignment from 500 Monte Carlo permutations. The solid line shows the sample mean of the coefficients and the dashed line indicates our real coefficients ($-19.50$). The probability of rejecting the null hypothesis is 0.088, indicating that the treatment effect is not a random result of sample selection.

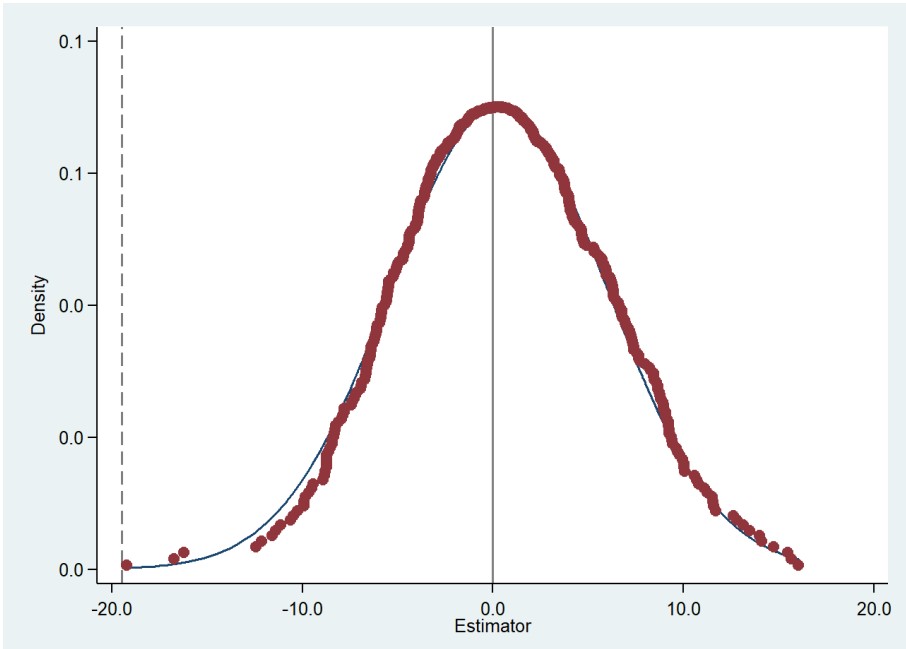

**Figure 6.** This figure shows the distribution of the estimated coefficients with randomly permuted assignment of censored weeks from 500 Monte Carlo permutations. The solid line shows the sample mean of the coefficients and the dashed line indicates our real coefficients ($-19.50$). The probability of rejecting the null hypothesis is 0.001, indicating that the treatment effect is not a random result of time trends.

## 6. Conclusions

While having many political and economic benefits, internet censorship is at the cost of economic productivity due to its segregation of cyberspace We exploit internet cen-

sorship intensity changes due to political events in China to study the impact of internet censorship on online labour production. With a novel dataset from a video game community, a difference-in-differences design finds that increased censorship intensity reduces online labour output by eight percent for volunteers in mainland China relative to those overseas. The reduction is mainly in the intensive margin where volunteers, while keeping the same working time, suffer from reduced work efficiency. However, we cannot rule out the extensive margin effect as the volunteers are already experienced in censorship bypassing. It is still possible for a general Chinese internet user to lose access to uncensored information during politically sensitive periods. Therefore, our result shows a lower bound of productivity loss due to the strengthened internet censorship intensity.

This paper extends beyond the literature to examine whether internet censorship reduces online labour work as an unwanted effect. As far as we know, this is the first study to shed light on the online labour productivity impact caused by internet censorship in China. A survey of the voluntary, recreation-oriented job might be generalized in a much broader online labour production issue. We understand the imperfection of our arguments, mainly from the difference between the nature of a real occupational job and volunteer work, and from the limitations in measuring GFW censorship intensity. However, the results suggest that online labour productivity losses may generate considerable economic costs when the government censors the internet. This should be carefully considered when making censorship policies.

**Author Contributions:** Conceptualization, J.F.; methodology, J.F.; software, J.F.; validation, J.F. and R.G.; formal analysis, J.F.; investigation, J.F. and R.G.; resources, J.F. and R.G.; data curation, J.F and R.G.; writing—original draft preparation, J.F.; writing—review and editing, J.F. and R.G.; visualization, J.F.; supervision, J.F.; project administration, J.F.; funding acquisition, J.F. and R.G. All authors have read and agreed to the published version of the manuscript.

**Funding:** This study receives no external funding support.

**Institutional Review Board Statement:** Not applicable.

**Informed Consent Statement:** Not applicable.

**Data Availability Statement:** The study used data from the "Shenzhen Enlightened" (Ingress Player Community), including a anonymous record of player review performance and a survey. Data are available upon request. The study used Stata 14.0 to process the data and analyse the results. Code with notes in English and Chinese Simplified is available upon request.

**Acknowledgments:** We thank Siying Qu (`@Davidia`) and another two anonymous Ingress players from the Shenzhen Enlightened community (their Ingress Agent IDs are `@Ethern` and `@Lokpro`) for offering the volunteer community records, `@immeIonji` for providing technical support regarding the GFW, and the participants of the CSAE IFS online meeting in February 2021 for thier helpful suggestions.

**Conflicts of Interest:** The authors declare no conflicts of interest.

## Appendix A. Survey Questionnaire

Note: The survey was carried out 6 months after the event. Here, the English version is shown. The original questionnaire was in Chinese.

Thank you for supporting this event! Your reply here will help us better serve future contestants.

**1. Which of the following descriptions best matches your motivation for signing up?**

– Incentives to accumulate a number of in-game achievements

– Contribute more Portals (including PokeStop, etc.) to your community

– In order to gain social recognition and self-satisfaction

– In order to win prizes

– For fun

**2. Which of the following descriptions best fits the reason you were out?**

– Due to personal reasons such as work, studies, etc.

– Unable to spare enough time to review nominations
– Forgot to upload data or forgot to open the profile page
– The network is unstable
– I have encountered frequent cooling issues
– I was not out and I finished the game

**3. Which of the following descriptions best fits your access to the OPR (Wayfarer) website during the competition?**
– I purchased a network proxy services from people I don't personally know
– I use a network proxy service built by myself or my friends
– I use the network agency service provided by my school or employer
– I don't need a special network proxy service to access the OPR/WFR website

**4. During the competition, how many hours did you spend on average reviewing each week?**
– 0–2 h
– 2–4 h
– 4–6 h
– 6–8 h
– More than 8 h
– Cannot remember

**5. During the competition, what was the maximum number of hours you spent per week reviewing?**
– 0–2 h
– 2–4 h
– 4–6 h
– 6–8 h
– More than 8 h
– Cannot remember

**6. During the competition, what the was the minimum number if hours you spent per week reviewing?**
– 0–2 h
– 2–4 h
– 4–6 h
– 6–8 h
– More than 8 h
– Cannot remember

**7. In the month AFTER the competition, how many hours did you spend on average reviewing each week?**
– 0–2 h
– 2–4 h
– 4–6 h
– 6–8 h
– More than 8 h
– Cannot remember

**8. Approximately how much time do you spend playing Ingress each week? OPR/Intel/decoding are not included.**
– 0–2 h
– 2–4 h
– 4–6 h
– 6–8 h
– More than 8 h
– Cannot remember

**9. What is your overall evaluation of this event?**
– Very satisfied
– Quite satisfied

– Neutral
– Not so satisfied
– Very dissatisfied
**10. How did you like the promotion and sign-in process?**
– Very satisfied
– Quite satisfied
– Neutral
– Not so satisfied
– Very dissatisfied
**11. How did you like our data-checking with [Agent-stats upload] + [volunteer spot check]?**
– Very satisfied
– Quite satisfied
– Neutral
– Not so satisfied
– Very dissatisfied
**12. If this event was held again, what is your willingness to participate?**
– Have a strong willingness to participate again
– Some willing, participation depends on other factors
– Would not participate again
**13. If this event was held again, do you have any intention of becoming a volunteer? You can also assist if you don't participate.**
– Yes
– No
**14. If this event was held again, how many weekly agreements do you think is appropriate?**
– 60
– 70
– 80
– 90
– 100
– Other (write-in)
**15. If this event was held again, how long do you think is appropriate for the whole competition?**
– Within 3 months
– 3 to 6 months
– More than 6 months
– Other (write-in)
**16. If this event was held again, what kinds of prizes are attractive to you?**
– Ingress-related virtual inventories, such as Very-Rare LOAD-OUT or Character Medals
– Ingress-related collections, such as official/fan stickers or badges
– Other virtual prizes, such as console game redemption codes
– Cash
– Other (write-in)
**17. Any other comments or suggestions?**
– No
– Other (write-in)
Thank you for your patience to fill in this questionnaire!

## Notes

[1] See the official website https://ingress.com/support accessed on 21 March 2023 for a detailed introduction to the portals.

[2] The initial portals are mostly derived from the Historical Marker Database (HMdb), containing less than a million.

[3] The exact decision algorithm is confidential, but mostly it depends on the number of reviewers and whether the reviewers have similar ratings.

[4] Some restrictions are observed during the politically sensitive periods; for example, the security check for public transportation may intensify in Beijing, but we believe the impact on people's mobility is subtle.

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
