# Peer review of "Estimating the Cost of Internet Censorship in China: Evidence from a Gamified Remote Platform"

_journalmedia, doi:10.3390/journalmedia4020027_

Round 1
Reviewer 1 Report
1. Abstract: the abstract must be rewritten according to the scientific methods
2. Introduction: The introduction must be rewritten in more details, to clarify the scientific gradient of the topic.
3. Background Information and Data: can be improved by providing convincing evidence to the reader.
4. The statistical analysis of the research is very weak, and no modern statistical method has been used.
5. The conclusions need to be strengthened and restructured, due to the dispersion of the theoretical part and the weakness of the practical part
Reviewer 2 Report
The article starts with the claim of analyzing the costs and merits of the internet censorship, however in the conclusion there isn’t any clear elaboration on this fact. One of the first question is the formulation of the bias between economic productivity and political stability. Is it the only bias? The author’s hypothesis on this bias should be expressed in not just words, but with some concepts from the relevant literature I believe. The aim of the article is not also clear or clearly presented.
If it is not only a data presentation and article of social science, I would suggest to include more literature and theoretical approach. At least, the author for instance should define political stability and economic productivity. (Economic productivity of whom? The individual firms or government and how it is measured). Moreover, from a political science perspective the merits and costs of censorship should be explained. Merit or cost from the perspective of human rights?, security? National unity?, civil rights?, liberty etc..
Round 2
Reviewer 1 Report
the authors did well
Reviewer 2 Report
It is good to see that the author now gives place to some relevant literature. The conclusion part is better and the argument of the article is clearer as presented in the conclusion. However, especially in terms of conceptualisation, the article needs to be further studied.